# Uptake of best practice recommendations in the management of patients with diabetes and periodontitis: a cross-sectional survey of healthcare professionals in primary care

Susan M Bissett ,[1] Tim Rapley ,[2] Philip M Preshaw,[3] Justin Presseau[4]

¹School of Dental Sciences, University of Newcastle upon Tyne, Newcastle upon Tyne, UK
²Social Work, Education and Community Wellbeing, Northumbria University Department of Social Work and Communities, Newcastle upon Tyne, UK
³National University Centre for Oral Health, National University of Singapore, Singapore
⁴School of Epidemiology and Public Health, and the School of Psychology, University of Ottawa, Ottawa, Ontario, Canada

**Correspondence to**
Dr Susan M Bissett;
s.m.bissett@newcastle.ac.uk

## ABSTRACT

**Objectives** To investigate the practices of healthcare professionals in relation to best practice recommendations for the multidisciplinary management of people with diabetes and periodontitis, focusing on two clinical behaviours: informing patients about the links between diabetes and periodontitis, and suggesting patients with poorly controlled diabetes go for a dental check-up.

**Design** Cross-sectional design utilising online questionnaires to assess self-reported performance and constructs from Social Cognitive Theory (SCT) and Normalisation Process Theory.

**Setting** Primary care medical practices (n=37) in North East, North Cumbria and South West of England Clinical Research Networks.

**Participants** 96 general practitioners (GPs), 48 nurses and 21 healthcare assistants (HCAs).

**Results** Participants reported little to no informing patients about the links between diabetes and periodontitis or suggesting that they go for a dental check-up. Regarding future intent, both GPs (7.60±3.38) and nurses (7.94±3.69) scored significantly higher than HCAs (4.29±5.07) for SCT proximal goals (intention) in relation to informing patients about the links (p<0.01); and nurses (8.56±3.12) scored significantly higher than HCAs (5.14±5.04) for suggesting patients go for a dental check-up (p<0.001). All professional groups agreed on the potential value of both behaviours, and nurses scored significantly higher than GPs for legitimation (conforms to perception of job role) in relation to informing (nurses 4.16±0.71; GPs 3.77±0.76) and suggesting (nurses 4.13±0.66; GPs 3.75±0.83) (both p<0.01). The covariate background information (OR=2.81; p=0.03) was statistically significant for informing patients about the links.

**Conclusions** Despite evidence-informed best practice recommendations, healthcare professionals currently report low levels of informing patients with diabetes about the links between diabetes and periodontitis and suggesting patients go for a dental check-up. However, healthcare professionals, particularly nurses, value these behaviours and consider them appropriate to their role. While knowledge of the evidence is important, future guidelines should consider different strategies to enable implementation of the delivery of healthcare interventions.

## Strengths and limitations of this study

► Although best practice recommendations for the multidisciplinary management of patients with diabetes and periodontitis have been published, there is limited research on their uptake; this is the first study to explore practices of UK medical healthcare professionals in the context of such recommendations.

► We achieved a high participation rate (80% at the practice level, and 76% at the participant level) from a range of healthcare professionals (general practitioners (GPs), nurses and healthcare assistants (HCAs)).

► It is acknowledged that generalisability may be limited given that participants were recruited by convenience sampling from two geographically separate locations in the UK (the North East and North Cumbria and South West Peninsula Clinical Research Networks) and the breakdown of GPs, nurses and HCAs who were sent the survey is unknown which means we cannot identify percentage response rate relative to job role.

► We utilised a novel combination of Social Cognitive Theory and Normalisation Process Theory to evaluate current practice at an individual and organisational level.

► The cross-sectional design precludes causal inference and direction of effect, but provides the first indication of factors associated with multidisciplinary management of people with diabetes and periodontitis in UK primary medical care.

## INTRODUCTION

Diabetes is a prevalent chronic noncommunicable disease that has significant impacts on well-being and quality of life. Type 1 and type 2 diabetes are the principal categories, and in the UK as of 2019, there were 3.8 million people diagnosed with diabetes, of whom around 90% have type 2 diabetes. In addition, almost 1 million people are thought to be living with type 2 diabetes that is not

BMJ

yet diagnosed. This brings the estimated total number of people affected by diabetes to 4.7 million, or approximately 1 in 10 of the UK population over 40 years of age. This figure is predicted to reach 5.5 million by 2030.[1] Treatment of diabetes requires a life-long management strategy and varies in complexity given the multifactorial aetiology of the condition.

Periodontitis is also a prevalent chronic non-communicable disease. It is characterised by bacterially-driven inflammation in the tooth-supporting tissues that results in connective tissue destruction and alveolar bone resorption, leading, ultimately to tooth loss.[2] The dental plaque biofilm initiates and perpetuates the inflammation and an early sign of gingival inflammation (gingivitis) is bleeding, for example after brushing the teeth. Periodontitis is generally slowly progressing and painless, thus patients commonly present with advanced periodontitis that they were hitherto unaware of or ignored, even if they had been aware of some bleeding. Severe periodontitis has been reported to be the sixth-most prevalent disease in the world[3] and prevalence data in the UK have shown 8% of adults have advanced periodontitis.[4]

Periodontitis was initially identified as being a complication of diabetes in the early 1990s, the risk of periodontitis being increased by two to three times in a person with poorly controlled diabetes compared with individuals without.[5 6] The pathogenic mechanisms linking periodontitis and diabetes are not yet fully understood but likely relate to upregulated systemic inflammation in each condition adversely affecting the other. The level of glycaemic control is key in determining risk,[7] and similar to the other complications of diabetes, the risk for periodontitis increases with poorer glycaemic control.[8–10]

Evidence supporting the potential to improve glycaemic control by treating periodontitis has emerged in more recent years, with several meta-analyses and two Cochrane reviews confirming reductions in glycated haemoglobin (HbA1c) following effective periodontal therapy of up to 3 to 4 mmol/mol (0.3% to 0.4%) 3 to 4 months after periodontal treatment.[11–14] The precise mechanisms for the reduction of HbA1c following periodontal treatment are not completely clear but likely arise from the combined effects of reduced inflammation and decreased bacterial challenge systemically, leading to a reduction in the systemic inflammatory state, and improvements in insulin resistance and insulin signalling.[10] Any reduction in HbA1c is important in people with diabetes, as it reduces the risk of diabetic complications. Every 1% reduction in HbA1c is associated with 21% reduced risk of any endpoint related to diabetes, 21% for deaths related to diabetes, 14% for myocardial infarction and 37% for microvascular complications.[15]

Over the years, a number of working groups have been established to provide guidance and recommendations for the multidisciplinary management of patients with periodontitis and diabetes. In 2007, the World Dental Federation and the International Diabetes Federation (IDF) jointly organised a symposium on Oral Health and Diabetes. Experts agreed that there was urgent need to inform professionals, people with diabetes, policymakers and the public about the impact of diabetes on oral health; and they produced the IDF Guideline on Oral Health for People with Diabetes.[16] This guideline recommends that healthcare professionals should advise people with diabetes that good oral hygiene and regular dental checks are important and, in addition, adequate oral hygiene should be considered a normal part of diabetes self-management. In 2012, experts from the European Federation of Periodontology (EFP) and American Academy of Periodontology also reviewed the evidence regarding the associations between diabetes and periodontitis.[17] They concluded that periodontitis was an independent predictor of several systemic conditions, including diabetes, and should be acknowledged as a major public health issue. Their manifesto stated that dental and medical communities should unite to develop a multidisciplinary approach to patient care.

Notwithstanding, previous research has shown that the academic and organisational silos in which dental and medical healthcare teams operate would appear to hinder shared knowledge and effective joint management of the two conditions.[18 19] As knowledge alone is not sufficient to enable implementation of the delivery of healthcare interventions, research to identify the investment potential of healthcare professionals and enablers of change to aid implementation would be valued.

Accordingly, this study investigated the reported practices of healthcare professionals in relation to the management of diabetes and periodontitis to ascertain whether published best practice recommendations[16 17] were being followed and to assess the factors which predict behaviour, focusing on two recommended clinical behaviours: (1) Informing patients with diabetes about the links between diabetes and periodontitis; and (2) suggesting patients with poorly controlled diabetes go for a dental check-up.

## METHODS
### Design
The study used a cross-sectional design, involving online questionnaires (Qualtrics) to collect healthcare professionals' self-reported performance and views on the two clinical behaviours. The Strengthening the Reporting of Observational Studies in Epidemiology cross-sectional reporting guidelines were followed.[20] As knowledge (or lack of knowledge) can influence cognitions towards behaviours, creating response bias,[21] we randomised (1:1 ratio) the provision (or not) of background information on the topic of the bidirectional relationship between diabetes and periodontitis as a preface at the start of the questionnaire, and the questionnaire was also piloted prior to use with healthcare professionals. As recommended by the UK Medical Research Council guidance for developing and evaluating complex interventions,[22] theory was used to explore healthcare professionals' behaviours at an individual and organisational level in

**Table 1** Definitions of social cognitive theory (SCT) and normalisation process theory (NPT) constructs utilised in this research

SCT: a theory of motivation and action that is used to predict healthcare professionals' cognitions that may improve quality of care. SCT comprises three constructs:

| | |
|---|---|
| Self-efficacy | The belief in one's ability to succeed in specific situations or accomplish a task. |
| Outcome expectations | One's expectations about the consequences of performing an action or behaviour. |
| Proximal goals | One's intention (ie, motivation) that regulates future effort and action with respect to a particular behaviour. |

NPT: a framework that is used to evaluate the factors that promote or inhibit implementation of processes (such as specific aspects of patient management) into routine care. NPT comprises four core constructs:

| | |
|---|---|
| Coherence | How healthcare professionals make sense of the behaviour or intervention, for example, what it involves and why? |
| Cognitive participation | How healthcare professionals get involved and stay committed, for example, can they see how they contribute? |
| Collective action | How healthcare professionals make it work in practice, for example, what do they need to make it happen? |
| Reflexive monitoring | How healthcare professionals assess whether it is worth the effort, for example, does it result in benefits to patient care? |

NPT also includes up to 16 sub-constructs, and those that are relevant to the particular clinical scenario should be selected. We selected five NPT sub-constructs in this research, and the participants were asked to respond to these in the questionnaire:

| | |
|---|---|
| Differentiation | I can see how the (behaviour) differs from usual ways of working. |
| Communal specification | Staff in this organisation have a shared understanding of the purpose of this (behaviour). |
| Individual specification | I understand how the (behaviour) affects the nature of my own work. |
| Internalisation | I can see the potential value of the (behaviour) for my work. |
| Legitimation | I believe that participating in the (behaviour) is a legitimate part of my role. |

Table adapted from Bandura (SCT),[23 24] May et al and Finch et al. (NPT).[25 26] Reproduced with permission from Bissett et al.[34]

the context of diabetes and periodontitis, specifically a combination of Social Cognitive Theory (SCT)[23 24] and Normalisation Process Theory (NPT).[25 26] A sample of the survey is included as a online supplementary file.

SCT is a theory of motivation and action that describes key modifiable cognitions that can help to explain and improve the quality of care.[27–29] SCT holds that the care provided by healthcare professionals is a function of their self-efficacy (belief in their ability to provide the care), their outcome expectations (beliefs about the consequences of the care they provide), their proximal goals (intention to provide the care) and the present sociostructural determinants (external social and structural factors that act as barriers and enablers to care provision). NPT is an implementation theory used to identify, conceptualise and evaluate the factors that promote or inhibit the introduction, implementation and embedding of processes (such as patient management) into normal care.[30 31] Subsequently, the NoMAD instrument[32 33] was developed as a tool for using NPT to quantitatively assess implementation determinants, comprising four core constructs: coherence, cognitive participation, collective action, reflexive monitoring[30] and sub-constructs or items. The NoMAD tool can be customised by selecting sub-constructs as appropriate according to the study context (table 1).[34]

## Measures

### Self-reported past behaviour

The questionnaire measured past behaviour in relation to the last 10 patients with diabetes seen for whom the healthcare professionals reported performing any of the two recommended clinical behaviours (informing and suggesting). Response options ranged from 0 to 10 patients (ie, the behaviour was performed on 'x' of their last 10 patients with diabetes) in order to simplify the estimation of that behaviour by the respondent, consistent with other studies that evaluated healthcare professionals' provision of diabetes-related healthcare.[35]

### SCT constructs

SCT constructs (self-efficacy, outcome expectations and proximal goals) were assessed using multi-item scales for both of the recommended clinical behaviours. Proximal goals was assessed on a 10-point scale to directly estimate for how many of the respondents' next 10 patients with diabetes they intended to engage in the behaviours. Self-efficacy and outcome expectations were also assessed for the two behaviours, using a 5-point Likert scale as follows: '1-strongly disagree', '2-disagree', '3-neither agree or disagree', '4-agree' and '5-strongly agree'. The wording of the items to assess SCT constructs was consistent with previous research.[35]

## NPT constructs

We customised the NoMAD tool to include five NPT sub-constructs: differentiation, communal specification, individual specification, internalisation and legitimation.[31] All were measured using the same 5-point Likert scale as shown above. Multi-item questions were informed by our previous qualitative research in the context of the management of people with periodontitis and diabetes.[19]

## Sample

Participants invited to complete the questionnaire included general practitioners (GPs), nurses and healthcare assistants (HCAs) working in primary care services involved in the care of people with diabetes. Participants were recruited via convenience sampling through the UK Clinical Research Network (CRN), specifically North East and North Cumbria (NENC) CRN and South West Peninsula (SWP) CRN. A network facilitator approached medical teams and gave them a study summary to consider. The number of practices that the CRN approached was not provided. Expressions of interest were emailed to the researcher, who scheduled a telephone call to obtain a contact list of the staff members who managed people with diabetes. An email invitation containing a link to the questionnaire was then sent to each staff member. Respondents were given 5 weeks to complete and submit the questionnaire. During this time, two electronic reminders were sent as these have been shown to improve response:[36] one at 3 weeks following the initial invitation, and the other after 4 weeks, thus reminding the respondent that there was only 1 week left before the close of the survey. Completion and submission of the questionnaire was incentivised at half of the respondent's professional hourly salary, given the typical time to complete established during the pilot phase. The questionnaire responses were anonymous. An a priori sample size target of n=150 was set, consistent with thresholds suggested in systematic reviews of studies using constructs from behaviour theories to predict medical professional behaviour.[27 37] The recruitment period ran from January 2016 to October 2016.

## Statistical analysis

Statistical analysis was conducted using SPSS V.23.0 for Windows. Summary sample characteristics and NPT data were calculated using descriptive analyses (means and SD). The internal consistency of multi-item constructs was tested in order to combine results to a single mean score.[38 39] Kruskal-Wallis tests were used to identify significant differences in responses according to professional role (GPs, nurses, HCAs), with Mann-Whitney post hoc tests with adjustment of the critical value of p as appropriate. SCT correlates of behaviour were assessed using binary univariate and multivariate logistic regression to identify construct predictors for each of the behaviours.

Site approval was granted from each practice principal who approached eligible staff members. Interested staff members were sent an email invitation to the survey and consented to participate by clicking the link and submitting the survey.

## Patient and public involvement

Patient and public involvement (PPI) was integrated throughout the life-cycle of the study, from funding proposal through to dissemination of results. The PPI group (the Oral and Dental PPI group of Newcastle University's School of Dental Sciences) contributed to the study design and development of the research question via meetings with the lead researcher, and there was PPI representation on the study steering committee which met at regular intervals during the research.

## RESULTS

The contact details of 46 medical practices were forwarded to the researcher by CRN facilitators: 11 from NENC CRN and 35 from SWP CRN. Of these, a total of 37 practices took part in the study (80% practice-level participation rate): 10 from NENC (27%) and 27 from SWP (73%). One hundred and seventy-six questionnaires were returned from 217 that were sent out. Reasons for non-participation and a breakdown of the job role of the 217 participants were not known. Partially completed questionnaires were deleted list-wise to achieve a final sample of 165: 96 GPs, 48 nurses and 21 HCAs (76% participant-level response rate). Sample sociodemographic and clinical practice descriptive statistics (table 2) show that the mean (±SD) number of patients with a diabetes diagnosis was 6.3%±2.4%, and 67.6% of practices operated a separate diabetes clinic. The majority of the respondents were female (72%). HCAs saw a greater mean number of patients with diabetes per month (37.7±40.8), compared with GPs (33.2±31.8) and nurses (29.7±26.0), although the SD were large.

## Behaviour 1: informing patients with diabetes about the links between diabetes and periodontitis

The questionnaire identified that healthcare professionals from all professional groups reported informing less than one of their last 10 patients with diabetes about the links between diabetes and periodontitis with no significant differences seen between GPs (0.23±0.69), nurses (0.58±1.81) and HCAs (0.24±0.63) (table 3). The GPs' mean self-efficacy score (2.82±0.76) did not differ from that of HCAs (2.94±0.82), but was significantly lower than that of nurses (3.19±0.76) (p=0.01). The barriers perceived to undermine self-efficacy to inform patients included 'it is not a priority for the patient' and 'I am running late', whereas the highest ranked self-efficacy undermining barrier by all three professional groups was 'there are problems accessing dental services', with healthcare professionals scoring positively that they would remain confident to inform patients despite this challenge. The GPs' mean score (3.10±0.74) for outcome expectations did not differ significantly from that of HCAs (3.38±0.74), but was significantly lower than that of

## Table 2 Sample characteristics of study population (n=165)

| Practice level characteristics (n=37) | | |
|---|---|---|
| Practice recruitment (N, %) | NENC | 10 (27%) |
| | SWP | 27 (73%) |
| List size (minimum-maximum) | | 3600–35 818 |
| Location (N, %) | Urban | 7 (18.9%) |
| | Rural | 11 (29.7%) |
| | Mixed | 19 (51.4%) |
| Practices with separate diabetes clinic (N, %) | | 25 (67.6%) |
| % patient list >65 years (mean±SD) | | 22.5%±6.4% |
| % patient list have diabetes diagnosis (mean±SD) | | 6.3%±2.4% |
| Sample level characteristics (n=165) | | |
| Sex (N, %) | Female | 119 (72.1%) |
| | Male | 46 (27.9%) |
| Age cohort (N, %) | <30 years | 5 (3.0%) |
| | 30–40 years | 39 (23.6%) |
| | 40–50 years | 58 (35.2%) |
| | 50–60 years | 56 (33.9%) |
| | >60 years | 7 (4.2%) |
| N patients with diabetes seen per month | GP (n=96) | 33.2±31.8 |
| | Nurse (n=48) | 29.7±26.0 |
| | HCA (n=21) | 37.7±40.8 |

Data for continuous variables presented as mean±SD.
%, percentage; GP, general practitioner; HCA, healthcare assistant; NENC, North East and North Cumbria; SWP, South West Peninsula.

nurses (3.54±0.90) (p=0.01). For proximal goals (intention to inform) the responses were generally positive with GPs' mean score (7.60±3.38) not differing significantly from that of nurses (7.94±3.69); however, they were both significantly higher than that of HCAs (4.29±5.07) (p=0.01).

The NPT responses from all professional groups for differentiation indicated that this behaviour differed from usual ways of working with the GPs' mean response (4.06±0.89) not differing significantly from that of nurses (3.91±1.00) or HCAs (3.67±1.05). GPs' mean response score to communal specification (staff in the organisation share understanding of the purpose of informing) (2.27±0.83) did not differ from that of nurses (2.63±1.09) or HCAs (2.86±1.10). There were no significant differences detected for individual specification between GPs (3.40±0.93), nurses (3.30±0.95) and HCAs (3.07±0.70). Responses for internalisation and legitimation were positive. For internalisation (or seeing the potential value of informing), GPs' mean score (4.08±0.66) did not differ from that of nurses (4.26±0.61) or HCAs (4.07±0.59). The last cognitive participation item, 'I believe informing

patients is a legitimate part of my role' was responded to positively. HCAs were the least positive (3.57±0.65) and significantly different to nurses (4.16±0.71), who also scored significantly higher than GPs (3.77±0.76) (p=0.01).

The SCT predictors for informing accounted for a small amount of variance (Cox & Snell $R^2$ 0.05; Nagelkerke $R^2$ 0.09) (table 4). The covariate background information/no background information (OR=2.81; p=0.03) was statistically significant, indicating that it does appear to be associated with responses about informing patients about the links. Neither self-efficacy (OR=1.07, p=0.82), outcome expectations (OR=1.49, p=0.21) or proximal goals (OR=1.10, p=0.21) were significant predictors in a model that controlled for demographic factors and included other SCT constructs.

### Behaviour 2: suggesting patients with poorly controlled diabetes go for a dental check-up

All three professional groups reported suggesting patients with poorly controlled diabetes go for a dental check-up to none or one of the last 10 of their patients, with HCAs reporting the least amount (0.14±0.48), followed by GPs (0.29±0.71) and then nurses (1.10±2.46) (table 3). The GPs' mean self-efficacy score (3.17±0.88) did not differ from that of HCAs (3.15±0.71), but was significantly lower than that of nurses (3.54±0.78) (p=0.01). The barriers that undermined self-efficacy to suggest patients go for a check-up most included 'it is not a priority for me/patient' and 'I am not set up for it', whereas the highest ranked item by all three professional groups was 'work being busy', with healthcare professionals scoring positively that they would remain confident to suggest patients with poorly controlled diabetes go for a check-up despite this challenge. GPs' mean score (3.24±0.80) for outcome expectations did not differ from that of HCAs (3.55±0.82) or nurses (3.58±0.88). For proximal goals (intention to suggest to go for a check-up), the responses were generally positive with the GPs' mean score (7.82±3.28) not differing from that of nurses (8.56±3.12), but the nurses' mean score was significantly higher than that of HCAs (5.14±5.04) (p<0.001).

The NPT responses for differentiation indicated that this behaviour was considered different from usual practice with GPs responding most strongly (4.14±0.63), but not significantly different to nurses (3.93±1.10) and HCAs (3.73±0.88). While the GPs' mean response score to 'staff in the organisation share understanding of the purpose of suggesting' (2.38±0.92) was statistically significantly lower than that of nurses (2.84±1.04) (p=0.01) and HCAs (3.27±1.03) (p<0.001), for individual specification, there was no significant difference detected between GPs (3.51±0.86), nurses (3.35±0.90) and HCAs (3.21±0.70). Responses for internalisation and legitimation were positive. Nurses' mean score (4.24±0.60) did not differ from that of GPs (4.06±0.65) or HCAs (3.87±0.52) for seeing the potential value of suggesting to go for a dental check-up. For legitimation, nurses' mean score (4.13±0.66) did not differ from that of HCAs (3.62±0.65),

**Table 3** Descriptive statistics of the two behaviours for SCT and NPT

| Behaviour | Job role | Past behaviour | Self-efficacy | Outcome expectations | Proximal goals | Differentiation | Communal specification | Individual specification | Internalisation | Legitimation |
|---|---|---|---|---|---|---|---|---|---|---|
| Informing | GP (n=96) | 0.23±0.69 | 2.82±0.76 | 3.10±0.74 | 7.60±3.38 | 4.06±0.89 | 2.27±0.83 | 3.40±0.93 | 4.08±0.66 | 3.77±0.76 |
| | Nurse (n=48) | 0.58±1.81 | 3.19±0.76 | 3.54±0.90 | 7.94±3.69 | 3.91±1.00 | 2.63±1.09 | 3.30±0.95 | 4.26±0.61 | 4.16±0.71 |
| | HCA (n=21) | 0.24±0.63 | 2.94±0.82 | 3.38±0.74 | 4.29±5.07 | 3.67±1.05 | 2.86±1.10 | 3.07±0.70 | 4.07±0.59 | 3.57±0.65 |
| | p | 0.95 | **0.04** (GP vs Nurse **0.01**) | **0.01** (GP vs Nurse **0.01**) | **0.01** (GP vs HCA **0.01**) (Nurse vs HCA **0.01**) | 0.29 | **0.03** | 0.25 | 0.32 | **0.01** (GP vs Nurse **0.01**) (Nurse vs HCA **0.01**) |
| Suggesting | GP (n=96) | 0.29±0.71 | 3.17±0.88 | 3.24±0.80 | 7.82±3.28 | 4.14±0.63 | 2.38±0.92 | 3.51±0.86 | 4.06±0.65 | 3.75±0.83 |
| | Nurse (n=48) | 1.10±2.46 | 3.54±0.78 | 3.58±0.88 | 8.56±3.12 | 3.93±1.10 | 2.84±1.04 | 3.35±0.90 | 4.24±0.60 | 4.13±0.66 |
| | HCA (n=21) | 0.14±0.48 | 3.15±0.71 | 3.55±0.82 | 5.14±5.04 | 3.73±0.88 | 3.27±1.03 | 3.21±0.70 | 3.87±0.52 | 3.62±0.65 |
| | p | 0.07 | **0.02** (GP vs Nurse **0.01**) | 0.81 | **<0.001** (Nurse vs HCA **<0.001**) | 0.25 | **<0.001** (GP vs Nurse **0.01**) (GP vs HCA **<0.001**) | 0.21 | 0.09 | **0.01** (GP vs Nurse **0.01**) |

Past behaviour and proximal goals were 10-point scales, that is, for how many of the last 10 patients does the clinician perform the behaviour ('past behaviour'), and for how many of their next 10 patients does the clinician plan to perform the behaviour ('proximal goals'); the other measures were 5-point Likert scales: '1-strongly disagree', '2-disagree', '3-neither agree or disagree', '4-agree' and '5-strongly agree'.

Data presented as mean±SD.

Statistically significant differences are indicated in bold font.

Post-hoc analysis for communal specification (informing) did not identify statistically significant differences between groups following adjustment of critical value of p.

P = test of differences between professional groups determined using Kruskal-Wallis (top level p value), with results of post hoc Mann-Whitney tests (with adjustment of critical value of p) presented in parentheses.

GP, general practitioner; HCA, healthcare assistant; NPT, normalisation process theory; SCT, social cognitive theory.

**Table 4** Multivariate logistic regression model predicting past informing and suggesting (n=165)

| Behaviours | Covariates and SCT predictors | OR | SE | P | 95% CI (B coefficient) | |
|---|---|---|---|---|---|---|
| | | | | | Lower | Upper |
| Informing* | Background/no background | 2.81 | 0.47 | **0.03** | 1.11 | 7.10 |
| | Self-efficacy | 1.07 | 0.30 | 0.82 | 0.60 | 1.92 |
| | Outcome expectations | 1.49 | 0.32 | 0.21 | 0.80 | 2.79 |
| | Proximal goals | 1.10 | 0.07 | 0.21 | 0.95 | 1.27 |
| Suggesting† | Background/no background | 1.26 | 0.40 | 0.57 | 0.58 | 2.74 |
| | Self-efficacy | 1.71 | 0.27 | **0.04** | 1.02 | 2.88 |
| | Outcome expectations | 1.06 | 0.26 | 0.81 | 0.64 | 1.75 |
| | Proximal goals | 0.99 | 0.06 | 0.90 | 0.88 | 1.12 |

P: Statistically significant predictors indicated in bold font.
*Cox & Snell $R^2$ 0.05, Nagelkerke $R^2$ 0.09.
†Cox & Snell $R^2$ 0.04, Nagelkerke $R^2$ 0.07.
B, exponential of β (OR); SCT, social cognitive theory; SE, standard error.

but was significantly higher than that of GPs (3.75±0.83) (p=0.01) for believing that suggesting patients go for a dental check-up was a legitimate part of their role.

The SCT predictors for suggesting patients go for a dental check-up accounted for a small amount of variance (Cox & Snell $R^2$ 0.04; Nagelkerke $R^2$ 0.07) (table 4). The covariate background information/no background information (OR=1.26; p=0.57) was not associated with suggesting to go for a check-up, and outcome expectations (OR=1.06, p=0.81) and proximal goals (OR=0.99, p=0.90) were not statistically significant. Self-efficacy accounted for a statistically significant amount of variability (OR=1.71, p=0.04).

## DISCUSSION

An abundance of evidence links periodontitis and diabetes, including multiple meta-analyses and two Cochrane reviews which have confirmed the benefits of periodontitis treatment on diabetes control. Adding to a number of recommendation documents for clinical practice that have been published over the last decade or so, the EFP and IDF held a joint workshop in 2017 and produced outputs which were published simultaneously in the Journal of Clinical Periodontology and Diabetes Research and Clinical Practice,[40 41] including guidelines for healthcare professionals, dental professionals and patients on the management of diabetes and periodontitis. Despite the existence of evidence and recommendations, the present research confirms that key clinical actions involved in enacting these recommendations are underperformed. The present research also identified modifiable correlates associated with variation in reported performance, suggesting targets for intervention to improve guideline recommendation uptake. Our cross-sectional survey investigated the quantitative self-reports of GPs, nurses and HCAs for two extant best practice clinical behaviours in relation to diabetes and periodontitis care published at the time of our research.[16 17] These

included informing patients about the links between periodontitis and diabetes and suggesting patients with poorly controlled diabetes go for a dental check-up.

While SCT and NPT have previously been used effectively in health promotion in the context of chronic disease management,[42–44] the combination of these behavioural (SCT) and implementation (NPT) theories represents a novel approach to investigate clinicians' practices and behavioural correlates. We have previously reported on the practices of dental clinicians in relation to the management of patients with diabetes and periodontitis, again using a combination of SCT and NPT.[34] We identified that according to self-reports, dental professionals are highly likely to inform their patients about the links between the two diseases and consider the impact of periodontitis treatment on glycaemic control, but there was very little evidence of dental clinicians contacting the patient's GP, with the respondents indicating that they would tend to communicate with the GP via the patient, as opposed to using direct communication or referral mechanisms. Based on the self-reported medical health-care professionals' responses in this study, there was little to no informing patients about the links between diabetes and periodontitis or suggesting that patients go for a dental check-up. All professional groups agreed that although both behaviours differed from usual ways of working (differentiation), there was potential value in the behaviours, indicated through positive responses for internalisation and intention (proximal goals).

For both of the investigated behaviours, nurses' responses were significantly higher than those of GPs and HCAs for self-efficacy; and for proximal goals, GPs' and nurses' scores were statistically significantly higher than those of HCAs. Regarding legitimation, nurses scored statistically significantly higher than GPs for informing patients about the links; and significantly higher than GPs and HCAs for suggesting patients go for a dental check-up. Perhaps these responses reflect the pivotal role

that a nurse often plays in patient education and health promotion in diabetes management.[45 46]

While we achieved a high participation rate (80% at the practice level, and 76% at the participant level) from a range of healthcare professionals, generalisability may be limited given that participants were recuited by convenience sampling. Furthermore, and the breakdown of GPs, nurses and HCAs who were sent the survey is unknown which means we cannot identify percentage response rate relative to job role.

Diabetes management can be complex and it is therefore no surprise that 'it is not a priority for me/patient' should be ranked as having the potential to undermine self-efficacy for both behaviours within this context, however problems associated with accessing NHS dental care and work being busy would not prevent the respondents from performing the behaviours. Previous research identified barriers to healthcare professionals talking to patients with diabetes about the oral complications of diabetes,[19 47] but the present study would appear to indicate some investment potential which has not been reported in the literature before and which could improve the implementation of guidance. The low levels of current behaviours may explain the small amount of variance seen with the SCT predictors. Notwithstanding, background information, while not a statistically significant predictor for suggesting to go for a check-up, was statistically significant for informing; and self-efficacy was a significant predictor for suggesting, and these should be considered as targets for training and implementation interventions. Future research could design an intervention to target these factors, which is then evaluated in a cluster-randomised trial to assess whether the intervention increased the target behaviours.

## CONCLUSION

Despite strong evidence and best practice recommendations, healthcare professionals report low levels of informing patients with diabetes about the links between diabetes and periodontitis and suggesting to go for a dental check-up. However, healthcare professionals, particularly nurses, value these behaviours and perceive them to be legitimate to their role as diabetes educators. Professional role appears to be an important factor when considering who, in the diabetes multidisciplinary team, is best aligned to speak to patients about diabetes and periodontitis.

**Acknowledgements** We thank the participants who generously gave their time and the North East and North Cumbria (NENC) and South West Peninsula (SWP) Clinical Research Networks for facilitating recruitment.

**Contributors** SMB, TR, PMP and JP contributed to the conception and design of the study. SMB collected the data, which was supervised by TR, PMP and JP. JP, TR and PMP were the major contributors in analysis and interpretation of the data and in writing the manuscript. SMB, TR, PMP and JP were involved in drafting the work, revising it critically for important intellectual content and the final approval of the version published. SMB, TR, PMP and JP agree to be accountable for all aspects of the work. Members of Newcastle School of Dental Sciences Patient and Public Involvment group contributed to the study design and development.

**Funding** This research was funded by a UK National Institute for Health Research Doctoral Research Fellowship (DRF-2014-07-023) awarded to SMB.

**Competing interests** None declared.

**Patient consent for publication** Not required.

**Ethics approval** North West-Greater Manchester West NHS Research Ethics Committee (16/NW/0030).

**Provenance and peer review** Not commissioned; externally peer reviewed.

**Data availability statement** All data relevant to the study are included in the article or uploaded as supplementary information. A PDF of the online survey has been made available as a supplementary file. Permission is required from the authors for re-use.

**ORCID iDs**
Susan M Bissett http://orcid.org/0000-0002-1997-3027
Tim Rapley http://orcid.org/0000-0003-4836-4279

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
