## [Reviewer comments · BMJ Open]

ARTICLE DETAILS

TITLE (PROVISIONAL)	Uptake of best practice recommendations in the management of patients with diabetes and periodontitis: a cross-sectional survey of healthcare professionals in primary care
AUTHORS	Bissett, Susan; Rapley, Tim; Preshaw, Philip; Presseau, Justin

VERSION 1 – REVIEW

REVIEWER	Martin S Lipsky Roseman University of Health Sciences USA
REVIEW RETURNED	19-Jul-2019

GENERAL COMMENTS	I think this paper is a nicely written paper that merits publications. I have only a few minor recommendations. I was not sure how the participants were selected. The authors say they were recruited from the UK Clinical Research Network but were all members invited or was it a convenience sample only those who agree participated? How do I know that they are similar to the larger group of GPs and nurses in the UK? For the coding of the comments, who coded them - one or more of the authors if more than one coded how do we know they coded the same? The stats seem fine to me but I recommended that if the other reviewer is not more statistically adept then me, maybe a stat review is appropriate or the authors can sign off that a biostat person reviewed. When the survey was piloted did they check for reproducibility? I do think the authors should address limitations. Such as the "N" is fairly small and also address that study examined providers in a single region on a single country. This is an internationally read journal and so I want to hear how the result apply to me or maybe it doesn't. Hope the authors find these comments helpful. Overall, it is a really nicely done study.
--

REVIEWER	Manabu Morita Okayama University Graduate School of Medicine, Dentistry and Pharmaceutical Sciences, Japan
REVIEW RETURNED	21-Jul-2019

GENERAL COMMENTS	RE: Uptake of best practice recommendations in the management of patients with diabetes and periodontitis: a cross-sectional mixed methods survey of healthcare professionals in primary care
---

	My concerns are as follows;  1. The sample size was calculated before starting the study? The number of participants is enough to get the conclusion? 2. I guess the sex distributions of general practitioners, nurses and healthcare assistants might be different, which affect the result. 3. It might be more useful to readers to mention how to improve the variables relating to SCT and NPT in the discussion section.
--	---

REVIEWER	Heike Schütze University of Wollongong, Australia
REVIEW RETURNED	22-Sep-2019

GENERAL COMMENTS	Thank you for the opportunity to review this paper which aims to investigate the healthcare provider practices of providing patients advice on: 1. The link between periodontitis and diabetes; and 2. Suggesting patients with poorly controlled diabetes receive a dental check-up. I was asked specifically to comment on the methods and analyses of this paper and hope that my comments are useful to the authors. This paper uses constructs from SCT and NPT to evaluate healthcare provider practices in relation to best practice guidelines. The use of these two theories to evaluate behaviour at the individual and organisational level is fitting. It would be useful to the reader to clarify further that the theories are measuring these different aspects. These are shown in Table 1, but to reader not familiar with the two theories it would be useful to have further clarification. The paper states it is a cross sectional mixed methods design using questionnaires. The quantitative component is strong. From the description provided on p8 lines 49-50, the qualitative component comprised a free- text box to provide further comment for qualitative analysis. From this description it appears that only one free text box was provided, and as such calling this study mixed methods design as opposed to a quantitative cross sectional study is somewhat misleading. Furthermore, there does not appear to be any 'mixing' of the results, with both results currently reported separately as quantitative and qualitative results. If the qualitative results are going to remain in the paper, then this study would appear to have employed multi-methods, not mixed methods, and the title and design should be updated to reflect this. Due to the very large quantitative component, in its current form, I feel the paper would be stronger if was presented as a cross-sectional quantitative study. However, if the authors wish to include a qualitative component, However, if the authors wish to include a qualitative component, it will require substantial additional detail as currently little rigour is demonstrated in the qualitative component:  • An example of the questions for the free-text box(es) would be useful to demonstrate their aim and scope • Provide a clear description whether there is only one free-text box or more. If the former, suggest calling this a quantitative study.
--

	 • The methods should include the epistemological view which underpins the research. • Who developed the code frame? • Was it based on the constructs being examined, the two behaviours being investigated, or something else? • What method of thematic analysis was used? • Were multiple coders used? If not, how was researcher bias balanced/accounted for? • If multiple coders were used, how were opposing codes/themes dealt with? • Were the data triangulated with the quantitative results? • The qualitative results would be better presented under themes with supporting quotes weaved in the text to support the theme. If this study really is a mixed methods study, then the qualitative results and quantitative results could be presented under each construct or behaviour. Other comments:  • Under the subheader 'Measures' in the methods section, there are four subheaders: 'Self-reported behaviour', 'SCT constructs', 'NPT constructs', and 'Sample'. The latter does not fit as a measure. Is self-reported behaviour part of the SCT constructs assessment or a separate assessment? An example of the questionnaire would be very useful here. • Under 'Sample' in the above section, the number of practices the CRNs provided details for would be useful • P9 line 3-4: reference to study summary - is this a practice version of a participant information sheet? • P9 line 8: who emailed the invitation to the staff? Was it the research team or someone in the practice? • P10 ethics: Suggest revising word choice "favourable ethical opinion" to "ethics approval was received" or something similar. • P10 ethics: Whether site approval was granted from each practice principal needs to be stated • P10 ethics: a statement of how participants were informed of the study and how consent was received is required. • P10 results: it would be useful to know what the breakdown of the 217 participants so the reader knows the % response from GPs, nurses and HCAs. • P10 results: provide reasons for non-participation or state this information is not known • P11 Table 2: This is currently data for 3 tables presented under one table. These data might be better presented as 2 separate cross tabs, one at the practice level and one at the sample level. The N patients per month could be presented in the text.
--	---

	 • P12 lines 17-19: are the references to, 'It is not a priority for the patient' and 'I am running late' from the survey? Again a sample of the questionnaire would be useful here. Considering Likert scales are used, an example of the questions for only SCT or NPT component may be sufficient? • P14 line 21: word choice - "one or less of the last 10 of their patients". Suggest "none or one of their last 10 patients". • P19 line 12: specify acronyms EFP and IDF. • P19 lines 52-59 through to p20 lines 3 -26: There is a lot of text devoted to explaining the authors previous publication. I wonder why this can't be summarised in 1-2 concise sentences to inform the following paragraph. • Limitations?? I wish the authors every success and look forward to the opportunity to review a revised manuscript.
--	--

VERSION 1 – AUTHOR RESPONSE

Reviewers' comments	Response
Reviewer 1	
1. I was not sure how the participants were selected. The authors say they were recruited from the UK Clinical Research Network but were all members invited or was it a convenience sample only those who agree participated? How do I know that they are similar to the larger group of GPs and nurses in the UK?	The participants were recruited via convenience sampling. The CRNs had a tiered approach to recruiting practices. They approached practices new to research initially as it was considered a simple study. They then approached gradually more experienced practices until the recruitment period closed. We have added a sentence in the 'Sample' paragraph in the Methods to clarify the convenience sampling, in addition to the limitations section.
2. For the coding of the comments, who coded them - one or more of the authors if more than one coded how do we know they coded the same? The stats seem fine to me but I recommended that if the other reviewer is not more statistically adept then me, maybe a stat review is appropriate or the authors can sign off that a biostat person reviewed. When the survey was piloted did they check for reproducibility?	The qualitative data have been removed from the paper, in view of comments from Reviewer 3 regarding the qualitative data (which was sourced from a free text box), and considerations as to whether the design was truly mixed-methods (as opposed to multi-methods). Regarding piloting of the survey, reproducibility was assessed as part of the think-aloud process, along with assessment of ease of navigation and functionality (and reproducibility) of the e-questionnaire software.
3. I do think the authors should address	We have added text in the limitations section

limitations. Such as the "N" is fairly small and also address that study examined providers in a single region on a single country. This is an internationally read journal and so I want to hear how the result apply to me or maybe it doesn't.	of the paper to highlight the issue of generalizability associated with the convenience sampling, but we pre-specified a sample size based on powering the analysis to detect an effect size that we assumed based on the literature and we exceeded that sample size.
Reviewer 2	
1. The sample size was calculated before starting the study? The number of participants is enough to get the conclusion?	An a priori sample size target of n=150 was set, consistent with thresholds suggested in systematic reviews of studies using constructs from behaviour theories to predict medical professional behaviour (already described in the paper). The final data set was over the target (n=165), and we feel that the number of participants is enough for the conclusions to be meaningful.
2. I guess the sex distributions of general practitioners, nurses and healthcare assistants might be different, which affect the result.	The primary objective of this study was to evaluate the practices of different healthcare professionals in relation to best practice recommendations on the management of patients with diabetes and periodontitis, and as such, it was not within the scope of the research to formally evaluate any impact of gender on the outcomes. We recognise that this may be a question to ask in future research, but our study was not powered or designed to evaluate this.
3. It might be more useful to readers to mention how to improve the variables relating to SCT and NPT in the discussion section.	A sentence has been added to the discussion: Future research could design an intervention to target these factors, then evaluated in a cluster-randomized trial to assess whether the intervention increased the target behaviours.
Reviewer 3	
1. This paper uses constructs from SCT and NPT to evaluate healthcare provider practices in relation to best practice guidelines. The use of these two theories to evaluate behaviour at the individual and organisational level is fitting. It would be useful to the reader to clarify further that the theories are measuring these different aspects. These are shown in Table 1, but to reader not familiar with the two theories it would be useful to have further clarification.	Text has been added to the strengths statement relating to SCT & NPT and to the design paragraph to clarify that the evaluation was at an individual and organisational level.
2. The paper states it is a cross sectional mixed methods design using questionnaires. The quantitative component is strong. From the description provided on p8 lines 49-50, the qualitative component comprised a free-text box to provide further comment for qualitative analysis. From this description it appears that only one free text box was provided, and as such calling this study mixed methods design as opposed to a quantitative cross sectional study is somewhat	Thank you for these helpful comments. We agree and have removed the qualitative elements from the manuscript. We have also modified the title of the paper accordingly. It now is presented as a cross-sectional quantitative study.

misleading. Furthermore, there does not appear to be any 'mixing' of the results, with both results currently reported separately as quantitative and qualitative results. If the qualitative results are going to remain in the paper, then this study would appear to have employed multi-methods, not mixed methods, and the title and design should be updated to reflect this. Due to the very large quantitative component, in its current form, I feel the paper would be stronger if was presented as a cross-sectional quantitative study.	
If the authors wish to include a qualitative component, it will require substantial additional detail as currently little rigour is demonstrated in the qualitative component:  •An example of the questions for the free-text box(es) would be useful to demonstrate their aim and scope •Provide a clear description whether there is only one free-text box or more. If the former, suggest calling this a quantitative study. •The methods should include the epistemological view which underpins the research. •Who developed the code frame? •Was it based on the constructs being examined, the two behaviours being investigated, or something else? •What method of thematic analysis was used? •Were multiple coders used? If not, how was researcher bias balanced/accounted for? •If multiple coders were used, how were opposing codes/themes dealt with? •Were the data triangulated with the quantitative results? •The qualitative results would be better presented under themes with supporting quotes weaved in the text to support the theme. If this study really is a mixed methods study, then the qualitative results and quantitative results could be presented under each construct or behaviour. 	We have amended the manuscript by removing the qualitative components, so the paper is now presented purely as a quantitative cross sectional survey.
Other comments:  •Under the subheader 'Measures' in the methods section, there are four subheaders: 'Self-reported behaviour', 'SCT constructs', 'NPT constructs', and 'Sample'. The latter does not fit as a measure. Is self-reported behaviour part of the SCT constructs assessment or a separate assessment? An example of the questionnaire would be very useful here. 	'Sample' is not a measure, so the sub-heading has been formatted bold to distinguish it.
 •Under 'Sample' in the above section, the number of practices the CRNs provided details for would be useful 	This information was not provided to us by the CRNs. Their method of working was to provide us only with information on the number of practices who were interested in

	participating, not how many were approached or contacted.
•P9 line 3-4: reference to study summary - is this a practice version of a participant informationsheet?	Yes, this is a one sheet summary which was produced for the study for use by the CRNs and approved by REC.
•P9 line 8: who emailed the invitation to the staff? Was it the research team or someone in the practice?	The research team entered the email addresses provided by the practice of anyone involved in diabetes care into qualtrics and the software sent a standardised email with a link.
•P10 ethics: Suggest revising word choice “favourable ethical opinion” to “ethics approval wasreceived” or something similar.	Amended.
•P10 ethics: Whether site approval was granted from each practice principal needs to be stated	Added.
• P10 ethics: a statement of how participants were informed of the study and how consent was received is required.	Site approval was granted from each practice principal who approached eligible staff members. Interested staff members were sent an email invitation to the survey and consented to participate by clicking the link and submitting the survey. Text has been added to the manuscript to further clarify this process.
• P10 results: it would be useful to know what the breakdown of the 217 participants so the reader knows the % response from GPs, nurses and HCAs.	We received 217 email addresses from practices for staff members who were involved in diabetes care. At that stage, we didn't know what the job role was – job roles were collected from submitted surveys of which there were 176. 11 of these were deleted list-wise and the final data set is 96 GPs, 48 nurses and 21 HCAs.
• P10 results: provide reasons for non-participation or state this information is not known	We have included a sentence indicating that the reason for non-participation was not known.
• P11 Table 2: This is currently data for 3 tables presented under one table. These data might be better presented as 2 separate cross tabs, one at the practice level and one at the sample level. The N patients per month could be presented in the text.	There are a number of different items of information presented in Table 2, at the practice level and the participant level, some of which are frequency data (N & percent values), some of which are continuous data (means and SD). We considered a number of ways to present the data, including in separate tables (one at the practice level, and one at the participant level), but even doing that would necessitate presenting different types of data (e.g. frequencies and continuous data) in the same tables. Overall, we feel that Table 2 is easy enough to understand, and there is value in having all of the information present in one table for ease of reading, so prefer to leave as is but if the reviewer feels strongly about this we are happy to reconsider.
• P12 lines 17-19: are the references to, 'It is not a priority for the patient' and 'I am running late' from the survey? Again a sample of the questionnaire would be useful here.	We have include the e-survey as a supplementary file and referred to it in the design section of methods.

Considering Likert scales are used, an example of the questions for only SCT or NPT component may be sufficient?	
• P14 line 21: word choice - "one or less of the last 10 of their patients". Suggest "none or one of their last 10 patients".	Changed to 'none to one'
• P19 line 12: specify acronyms EFP and IDF.	These are specified in the introduction, but I have specified them again here for the reader's convenience.
• P19 lines 52-59 through to p20 lines 3 -26: There is a lot of text devoted to explaining the authors previous publication. I wonder why this can't be summarised in 1-2 concise sentences to inform the following paragraph.	Agreed and text reduced.
• Limitations	Added

VERSION 2 – REVIEW

REVIEWER	Martin S Lipsky Roseman University of Health Sciences
REVIEW RETURNED	07-Oct-2019

GENERAL COMMENTS	I did not have the opportunity to see how the authors addressed the suggested concerns of the manuscript nor to see those of the other reviewer. However, as I recall my suggestions the authors seemed to have addressed them.
---

REVIEWER	Heike Schutze University of Wollongong, Australia
REVIEW RETURNED	27-Oct-2019

GENERAL COMMENTS	Thank you for inviting me to review the changes to this paper in response to my initial review. I feel the paper presents a lot better as a quantitative paper. The authors have responded adequately to reviewers comments. I have a couple of very minor points remaining: Sample – the authors have stated that the number of practices the CRNs approached was not provided. This would be useful in the text for the reader. Results: In response to the comment that it would be useful to know what the breakdown of the 217 participants so the reader knows the % response from GPs, nurses and HCAs the authors responded that they received this information in the survey. This information would be good to see in the analysis or at the very least recognise it as a limitation. I wish the authors every success.
--

VERSION 2 – AUTHOR RESPONSE

Reviewers' comments	Response
Reviewer 1	
Please note this reviewer was subsequently contacted with a further copy of your point-by-point response to the reviewers' comments and they remarked that the responses looked detailed and had addressed all reviewer concerns	Many thanks.
Reviewer 2	
I did not have the opportunity to see how the authors addressed the suggested concerns of the manuscript nor to see those of the other reviewer. However, as I recall my suggestions the authors seemed to have addressed them.	Many thanks.
Reviewer 3	
Thank you for inviting me to review the changes to this paper in response to my initial review. I feel the paper presents a lot better as a quantitative paper. The authors have responded adequately to reviewers comments. I have a couple of very minor points remaining:	Many thanks and responses below.
Sample – the authors have stated that the number of practices the CRNs approached was not provided. This would be useful in the text for the reader.	“The number of practices that the CRN approached was not provided.” This sentence has been added to the Sample paragraph.
Results: In response to the comment that it would be useful to know what the breakdown of the 217 participants so the reader knows the % response from GPs, nurses and HCAs the authors responded that they received this information in the survey. This information would be good to see in the analysis or at the very least recognise it as a limitation.	We have acknowledged that the breakdown of 217 participants is unknown in the results and added the following as a limitation: “Furthermore, the breakdown of GPs, nurses and HCAs who were sent the survey is unknown which means we cannot identify percentage response rate relative to job role.”